# Cascaded Sematic Segmentation for Kidney and Masses

Shi Ruize, Han Bing

Canon Medical Systems (China) Co., LTD
`ruize1.shi@cn.medical.canon`

**Abstract.** The segmentation for kidney and related masses is meaningful for the clinical surgery and treatment. In this paper, we use a cascaded method to locate the kidney area and segment the kidney as well as masses accurately based on 3D U-Net architecture. The region based segmentation is used in the training process to make result consistent with the evaluation criteria. The test result with Dice value is 0.9736, 0.8416 and 0.7971 for Kidney + Masses, Masses and Tumor respectively.

**Keywords:** Cascaded, U-Net, Region based

## 1 Introduction

Auto sematic segmentation performs well on organ segmentations in the last few years. Sematic segmentation for kidney and masses play an important role in the kidney related treatments such as ablation planning. Some CNN based net frameworks like nnUNet lead in some segmentation challenges and have promising performance in the product development. In this challenge, the U-Net framework is adopted as the backbone and combined with some method in kidney location, loss function, regions segmentation and segmentation steps in cascaded segmentation method.

## 2 Methods

### 2.1 Training and Validation Data

As for the dataset for training and validation, we select the ground truth which prefix "aggregated_MAJ" officially recommended, and no additional corrections were made to the ground truth. In the cascaded experiment, the kidney ROI regions datasets are cropped using the ground truth with the kidney regions connected domain's bounding box.

### 2.2 Preprocessing

In the preprocess, the training dataset intensity information is collected and analyzed. The HU values of each image data are clipped into 0.5 to 99.5 percent of the all foreground voxels. Every image is resampled into median spacing of all images, which is 3 mm, 0.78125 mm and 0.78125 mm in Z, X and Y axis respectively. To improve the diversity of training data and robustness of the trained model, some data augmentation process is used, such as random rotation, Gamma transform, Elastic transform and so on.

### 2.3 Proposed Method

We use the U-Net as the network architecture, the block information and some parameter information as shown in the Figure 1. To improve the accuracy of the segmentation result, we use the cascade network in this challenge, in the first stage, the lower resolution network is used to get the rough segmentation result for the kidney region, the second stage to get the high accuracy segmentation result of tumor and cyst from the kidney region ROI, as shown in the Figure 2. In the inference process for single image, the kidney region bounding boxes are obtained in the first stage. Cropping the input image with the bounding boxes to get the ROI regions as the inputs for the second stage. Merging the outputs of the second stage using the bounding box information got in stage one to get the final output of the

network.

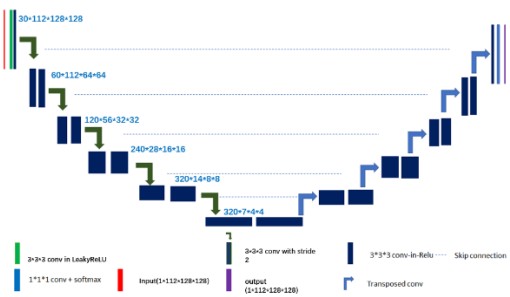

Figure 1 U-Net architecture and block parameters

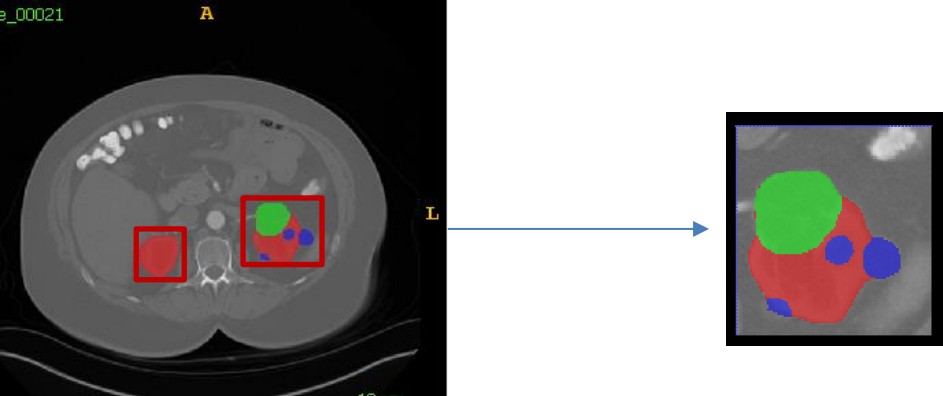

Figure 2 Cascaded segmentation for kidney and masses

In the common segmentation example, the kidney, tumor and cyst are treat with in tree channels separately, while the HECs(Hierarchical Evaluation Classes) is used in the evaluation process. In order to align the training process with the evaluation criteria, the region transform process is used in the data augmentation and compose to "Kidney and Masses: 0, 1, 2", "Masses: 1, 2", "Tumor: 1". As for the output of the network is shown as Figure 3.

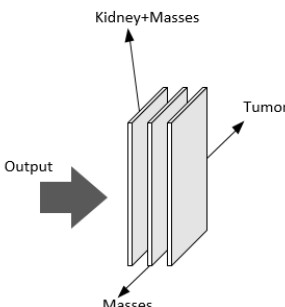

Figure 3 Network output for Region based segmentation

When the region based segmentation is adopted, it is not appropriate to use Softmax operation on the network output, here the Sigmoid function is used on it. As for the loss function, we use Dice combined with BCE.

The Batch size in default setting of the 3D nnUNet is 2 and the loss function based on sample batch Dice. In our

experiment, the Batch size is increased to 5 and modify the loss based on batch Dice, the result confirms that increasing Batch size can learn more volume information and perform well.

To improve the segmentation result, we reference to some tricks in the SOTA performance of other segmentation challenges and add them in our experiment such as increasing batch size, using batch dice in loss and region based segmentation.

## 3  Results

We do experiment in many groups and stages and compare with the nnUNet default setting. About the data set, I separate 60 sets from all data set as the test data, while not use the 5-folds validation mode. Table 1 show the experiment results using the Dice as evaluation metrics.

Table 1 Experiment results

|  | Kidney + Masses | Masses | Tumor | Average |
|---|---|---|---|---|
| Region based method | 0.9491 | 0.7338 | 0.6943 | 0.7924 |
| Default nnUNet | 0.9526 | 0.7588 | 0.7327 | 0.8147 |
| Cascaded Method | 0.9695 | 0.8367 | 0.7856 | 0.8639 |
| Region based model + BL+BD | 0.9589 | 0.7839 | 0.7625 | 0.8351 |
| Default nnUNet + BL+BD | 0.9702 | 0.8211 | 0.8057 | 0.8657 |
| Cascaded Method + BL +BD | **0.9736** | **0.8416** | **0.7971** | **0.8707** |

In the table, Region based method is the output the same as the evaluation criteria as shown in the Figure 3, Default nnUNet is the using the default setting of nnUNet. BL represents increasing batch size, BD means using batch dice in loss function.

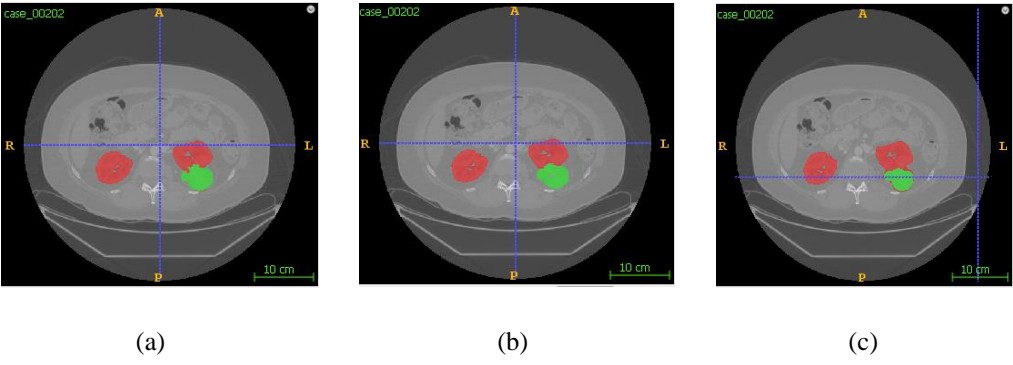

(a)                                      (b)                                      (c)

Figure 4 Segmentation results comparison

(a) Cascaded Method (b) Cascaded Method + BL +BD (c) Ground truth

## 3  Discussion and Conclusion

In the challenge, we use the U-Net as the as the backbone and refer to the practical experience of nnUNet. A cascaded segmentation method is adopted to segment the kidney ROI in rough way and segment the Masses accurately. Some tricks and optimized methods are used in experiments: (1) regions based segmentation (2) Batch size is increased (3) Batch based loss (4) Some data augmentation.

In my opinion, for the data set, label data is unevenly distributed especially for the tumor and cyst. If more data can be provided, then we can get a more robust and better result. For the limit time line, there are also some experiment that I want to do such as more data augmentation, more post process, these are not major improvements, but should

improve the evaluation results.

## Acknowledgements

I would like to express our gratitude to the organization of the KiTS21, it is a great challenge and learn much from it. Thanks to the contributors of nnUNet framework, it is very convenient for us to do experiments on our will.

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
