# OpenReview forum: "Cascaded Sematic Segmentation for Kidney and Masses"
_MICCAI.org/2021/Challenge/KiTS — Submitted to KiTS21 Challenge_

### Official Review · Reviewer_H7z4 · 2021-08-30

**Rating:** 5

**Review:**

The authors implemented six approaches, mostly centered around the nnU-Net baseline, and benchmarked each of them on a holdout set. The one with the highest dice performance was chosen for the final submission. In their case, this was a cascaded method with an increased batch size as well as batch-dependent dice loss. The authors do an adequate job explaining their method but the paper would benefit from an expanded narrative in various sections. In addition, the authors should do a better job at providing relevant context from the prior literature for their approaches and citing references to support their claims.

One crucial detail which is missing is how the multiple annotations per instance were aggregated together to produce a composite segmentation that could be used for training and validation. Most teams used majority voting as provided in the GitHub repository (aggregated_MAJ). If this is the case then it should be stated explicitly.

---

### Official Review · Reviewer_k1Y4 · 2021-08-30

**Rating:** 6

**Review:**

### Overall

- Please use the capitalization "KiTS21" where applicable

### Introduction

- It would be nice to expand this section. Perhaps you could briefly introduce your approach here before explaining it in detail. You could also provide a summary of what the rest of the paper entails.

### Methods

- Please expand on the statement "we have no effort to deal with de dataset in annotation"
- You mention that every image is sampled to median spacing. What are the actual numbers here?
- Instead of "Pic" please say "Figure"

### Results

- It would be nice to include a figure that shows an example prediction by your method and compares it with the corresponding ground truth
- It can be helpful to boldface the value in each column that represents the best performance

### Discussion and Conclusion

- Please try to stay away from phrases like "and so on". It's too casual for a research article such as this

---

### Decision · Program_Chairs · 2021-08-30

**Decision:**

Major Revisions

**Comment:**

Please address the reviewer comments and resubmit